# Evaluation of CRC-Metastatic Hepatic Lesion Chemoembolization with Irinotecan-Loaded Microspheres, According to the Site of Embolization

**DOI:** 10.3390/jpm12030414

**Published:** 2022-03-07

**Authors:** Marcin Szemitko, Elzbieta Golubinska-Szemitko, Marcin Warakomski, Aleksander Falkowski

**Affiliations:** 1Department of Interventional Radiology, Pomeranian Medical University, 70-111 Szczecin, Poland; zrz@pum.edu.pl; 2Department of General and Dental Diagnostic Imaging, Pomeranian Medical University, 70-111 Szczecin, Poland; e.golubinska@gmail.com; 3Department of General and Transplant Surgery, Pomeranian Medical University, 70-111 Szczecin, Poland; mkgwar@gmail.com

**Keywords:** colorectal cancer, metastases, liver chemoembolization, DEB-TACE, irinotecan

## Abstract

With the chemembolization of colorectal-cancer (CRC)-metastatic hepatic lesions by irinotecan-loaded microspheres, most researchers recommend slow embolizate delivery at the lobar-artery level to the entire liver parenchyma without obtaining visible stasis. An association has been reported between postoperatively visible embolizate stasis and lesion response to treatment. Possibly, in some cases, more selective administration might give greater benefit, particularly with previous systemic chemotherapy failure. Objective: Treatment response evaluation after chemoembolization of CRC-metastatic liver lesions with irinotecan-loaded microspheres, according to a hepatic-artery branch level of administration. Patients and methods: The analysis included 54 patients (24 females, 30 males) with large (median diameter > 5 cm) CRC-metastatic liver lesions, who underwent 196 chemoembolization procedures (mean 3.63 per patient) with irinotecan (100 mg)-loaded microspheres. Patients were divided into two groups according to initial embolizate-administration branch level: Group A (*n* = 26): at the segmental or subsegmental-vessel level; Group B (*n* = 28): at the lobar-branch level. Treatment response was assessed by computed-tomography (mRECIST criteria); overall survival (OS) and progression-free survival (PFS) were calculated using the Kaplan–Meier method and adverse effects were assessed according to the Common Terminology Criteria for Adverse Events (CTCAE; version 5.0). Results: There were statistically significant differences in the occurrence of partial response (PR): higher in Group A (42.3%) than Group B (17.9%) (*p* = 0.039) and occurrence of stable disease (SD): lower (*p* = 0.025) in Group A (11.5%) than Group B (39.4%). However, occurrence of disease progression (PD) was similar: Group A: 42.3%; Group B: 42.9% (*p* = 0.93). Patients in Group A presented with more favorable PFS (*p* = 0.029) and OS (*p* = 0.039) than Group B. Median survival times: Group A: 15.2 months; Group B: 13.1 months. There was no significant difference in complication incidence between groups (Group A: seven complications; Group B: six complications; *p* = 0.863). Conclusion: Superselective chemoembolizate administration to vessels supplying large CRC-metastatic liver lesions gave better response to treatment and extended patient survival time, without significantly increasing complication risk.

## 1. Introduction

More than half of patients diagnosed with colorectal cancer (CRC) develop hepatic metastases (CRHM), with surgical resection being the most effective method of treatment [1]. However, due to the involvement of liver parenchyma and the patients’ general condition, this is only possible in around 10–15% of patients [2]. Other treatment strategies for metastatic CRC include neoadjuvant and adjuvant chemotherapy, portal vein embolization, radiation therapy, thermal ablation, transarterial radioembolization (TARE) and chemoembolization (TACE). In patients who are ineligible for surgery, standard first-line chemotherapy is 5-fluorouracil (5-FU), folinic acid with either irinotecan (FOLFIRI) or oxaliplatin (FOLFOX), alone or with vascular endothelial growth factor (VEGF) inhibitors [3].

Liver chemoembolization can be used, alone or in addition to standard chemotherapy, and this often seems more effective than systemic chemotherapy [4]. Liver chemoembolization allows for high-dose delivery of the chemotherapeutic agent directly to the lesion, whilst reducing systemic exposure and the incidence of side effects due to irinotecan. Most researchers recommend slow delivery of embolizate at the branch level of the lobar arteries [5]. This delivers irinotecan microspheres to the entire liver parenchyma without obtaining embolizate stasis in the hepatic artery branches.

There are few studies which have reported using selective chemoembolization. Some recent studies have suggested the effect of tumor retention of contrast mixture and irinotecan-loaded microspheres on treatment response [6]. The chemoembolization of large lesions is particularly questionable, as patients often show a worse response to treatment, especially after prior systemic chemotherapy. For this reason, we have proposed a modification to the method of embolizate administration in cases of large metastatic lesions: initial superselective administration into the sub- and segmental arteries supplying the lesion with the creation of temporary stasis, followed by administration of the remaining embolizate at the level of the lobar arteries without the creation of stasis (Figure 1).

Although some authors have considered stasis formation as an end point of chemoembolization or have obtained stasis unintentionally [7], there are no studies in the literature assessing the efficacy of drug-eluting bead transarterial chemoembolization (DEB-TACE) with irinotecan according to the branch level of embolizate administration and the stasis formed.

The aim of the present study was to assess the relationship between the administration site of irinotecan microspheres (the branch level of embolization) and efficacy and safety in the treatment of liver metastatic lesions from colorectal cancer.

## 2. Materials and Methods

This retrospective study evaluated palliative chemoembolization procedures for unsuitable for surgery hepatic metastatic lesions from CRC, performed between November 2016 and December 2019. Patients (*n* = 54; 24 females, 30 males) with up to 10 metastatic lesions, with a mean diameter of the lesion greater than 5 cm, were analyzed. All patients had previously received FOLFIRI chemotherapy (with calcium folinate, 5-fluorouracil and irinotecan), and most had also received FOLFOX (folinic acid, 5-fluorouracil, and oxaliplatin) treatment.

A total of 196 chemoembolization procedures were performed using microspheres loaded with the cytostatic drug irinotecan (100 mg). The study was authorized by the Bioethics Committee at Pomeranian Medical University, Szczecin, Poland and all patients gave written informed consent.

Qualification for the procedure was obtained on the basis of abdominal computed tomography (CT) and/or magnetic resonance imaging (MRI) and laboratory results after consultation with a specialist oncologist, following European Society of Medical Oncology (ESMO) recommendations concerning the presence of advanced disease unsuitable for surgery due to the location of liver lesions and/or comorbidities, plus resistance to prior systemic therapy. Indications for treatment and inclusion into the study included: presence of hepatic lesions from metastasis with progression after prior systemic chemotherapy; Eastern Cooperative Oncology Group (ECOG) performance status ≤ 1; no evidence of liver failure; and age over 18 years old.

The exclusion criteria were: ECOG ≥ 2, liver failure, ascites, bilirubin level > 3 mg/dL, involvement of more than 50% of liver parenchyma, renal failure (creatinine > 2 mg/dL), thrombocytopenia (platelets < 50,000/mcl), and an allergy to contrast.

Patients were divided into two groups for different initial embolizate administration: Group A (*n* = 26), with the site of embolizate administration at the level of segmental or subsegmental vessels, and Group B (*n* = 28), in which embolizate was administered only via lobar branches. Response to treatment was assessed by means of a CT scan according to the modified Response Evaluation Criteria in Solid Tumors (mRECIST) criteria.

Treatment was performed according to a schedule of four procedures (or two if only one lobe of the liver was involved) at 3-weekly intervals, with alternating embolization administration via branches of the right or left hepatic artery and additional arteries supplying liver lesions. Microspheres (Embozene Tandem 100 µm microspheres; CeloNova Biosciences, now Varian Medical Systems, Inc, Palo Alto, CA, USA) were used. After irinotecan was loaded onto the microspheres, the supernatant was removed and the loaded microspheres were mixed with 10 mL of contrast agent (Iodixanolum 320 mg I/mL, GE Healthcare Inc, Marlborough, MA, USA).

The procedures were performed by interventional radiologists with skills certification and at least seven years of experience. Each patient received prophylactic antibiotics, steroids and proton pump inhibitors on the day before and on the day of the procedure, and additionally an antiemetic drug and an infusion of 1000 mL of 0.9% NaCl, ordered by the anesthesiologist attending the procedure, according to hospital guidelines.

### 2.1. Procedure

Chemoembolizate was administered using the Seldinger puncture method via the right or left common femoral artery. The coeliac trunk was then catheterized (in cases with hepatic artery anatomical variant visible from previous CT scan: other visceral arteries were also catheterized) using a SIM 5F catheter (Cordis/Johnson & Johnson, Miami, FL, USA), and arteriography and cone-beam CT were performed.

Hepatic vascularization and metastatic lesions were then assessed and, depending on their location and the image of the supply vessels, hepatic artery branches were catheterized using Progreat^®^ 2.7F microcatheters (Terumo, Tokyo, Japan). Patients in Group A were initially catheterized superselectively, starting with a catheterizable subsegmental or segmental branch supplying a particular tumor lesion visible via cone-beam CT. Administration of microspheres was preceded by injection of 2–3 mL of lidocaine into the catheter. Subsequently, under fluoroscopy, the mixture of microspheres and contrast agent was slowly injected (at a rate of approximately 1 mL/min), taking care that there was no reflux proximal to the catheter tip. The embolizate was administered until “near-stasis” (a stasis that resolves within seconds) was obtained at the level of the selectively catheterized subsegmental or, less commonly, segmental branch.

The above procedure was repeated in every possible superselective catheterized branch supplying the tumor lesion in question. Up to 50% of the previously prepared embolizate was superselectively injected. Branches that did not supply neoplastic lesions (as seen by cone-beam CT) were not catheterized. Procedures were terminated after administration of the remaining half of the embolizate with the microcatheter tip in the lobar artery, with the goal of delivering the embolizate to the entire treated region of the liver.

In patients in Group B, only the lobar artery was catheterized and embolizate was injected until all the drug was administered or until visible “near-stasis” was obtained at the level of small tumor-feeding vessels, with no stasis within sub- or segmental vessels.

During the procedure, the patient remained under the care of an anesthesiologist. Pain, during and after the procedure, was controlled with continuous infusion of opioids (20 mg morphine per day) and non-steroidal anti-inflammatory agents. Antiemetics (ondansetron 8 mg i.v.), dexamethasone (8 mg i.v.) and an antibiotic (cefazolin 1 g i.v.) were administered prophylactically twice daily. Most patients were discharged from hospital the day after the procedure.

### 2.2. Assessment of Complications

Complications were assessed on the basis of observations of the patient during hospitalization and at follow-up examinations at seven and 21 days after the procedure. Adverse effects and complications occurring perioperatively and within 30 days of the procedure were assessed according to the standards and terminology of the Common Terminology Criteria for Adverse Events (CTCAE; version 5.0). Data were saved in a form suitable for statistical evaluation (Excel 2007; Microsoft, Washington, DC, USA).

### 2.3. Feasibility of Chemoembolisation

A total of 196 chemoembolization procedures were performed in 54 patients. In 10 patients with single-lobe involvement, 20 chemoembolization procedures were performed. In the remaining 44 patients with two lobes involved, 176 chemoembolization procedures were performed. The technical success rate was 100% (Table 1).

### 2.4. Imaging and Tumor Response

Prior to, and one month after, the last procedure, an image from a triple-phase computed tomography scan or a contrast-enhanced magnetic resonance image was obtained to assess response. Response was assessed using the modified Response Evaluation Criteria in Solid Tumor (mRECIST); complete response (CR) was defined as disappearance of any intratumoral arterial enhancement in all target lesions; partial response (PR) was defined as at least a 30% decrease in the sum of diameters of viable enhancement in the arterial phase target lesions; progressive disease (PD) was defined as an increase of at least 20% in the sum of the diameters of viable enhancing target lesions and stable disease; and SD was defined as any cases that did not qualify for either partial response or progressive disease.

### 2.5. Statistical Analyses

Descriptive statistics of the studied variables were given as arithmetic means and standard deviations or as medians and ranges. Qualitative variables were analyzed using Pearson’s chi-squared tests. Continuous variables were compared using Student’s t tests or Mann–Whitney U tests for non-normally distributed variables. The relationships between treatment and the occurrence of complications were assessed using Pearson’s chi-squared tests. Overall survival (OS) and progression-free survival (PFS) analysis were performed using the Kaplan–Meier method and long-rank test. A *p*-value of <0.05 was considered significant. Calculations used commercial software Statistica ver. 13.1; www.statsoft.pl, accessed on 4 December 2021; StatSoft Polska, Krakow, Poland).

## 3. Results

### 3.1. Baseline Characteristics

Patient baseline characteristics showed no statistically significant differences between groups (Table 2). The mean diameter of the largest lesion was 9.4 ± 2.8 cm in Group A and 8.9 ± 2.4 cm in Group B. The mean number of hepatic metastatic lesions was 5.4 (range 1–10) in Group A and 5.7 (range 1–9) in Group B. The mean diameter of all lesions was 5.8 cm in Group A and 5.5 in Group B. Other patient characteristics are shown in Table 2.

### 3.2. Radiological Response after TACE

Comparison of treatment response between the two groups.

There was a statistically significant difference (*p* = 0.039) in the occurrence of partial response (PR), being higher in Group A (42.3%) compared to Group B (17.9%). The occurrence of stable disease (SD) was significantly higher (*p* = 0.025) in Group B (39.4%) than in Group A (11.5%). However, the occurrence of disease progression (PD) was similar in Group A (42.3%) and Group B (42.9%) (*p* = 0.93). An overall comparison of treatment response is shown in Figure 2. Complete remission (CR) was observed in one patient (3.8%) in Group A, from a single metastatic lesion.

### 3.3. Comparison of Progresion-Free Survival and Survival Times

After treatment, patients in group A presented a more favorable progression-free survival time (PFS) compared with group B (*p* = 0.029). The median PFS was 5.9 months in group A and 4.2 months in group B (Figure 3).

Survival time (OS) was significantly longer in Group A compared to Group B (*p* = 0.039). Median OS time was 15.2 months in Group A and 13.1 months in Group B (Figure 4).

### 3.4. Comparison of Adverse Events

There were a total of 13 (6.6%) significant complications in the chemoembolizations performed: 7 in Group A and 6 in Group B (not significantly different; *p* = 0.863).

In two patients (one in Group A and one in Group B), an anaphylactic reaction occurred during the procedure, which was resolved after an intervention by the anesthesia team. One patient in Group A experienced a septic episode with a liver abscess two weeks after the last procedure, successfully treated with antibiotic therapy. Three patients (including one from Group A) showed ultrasound signs of cholecystitis, which was resolved after conservative treatment. Two patients in Group A had concomitant occlusion of the right and left branch of the hepatic artery. In another two patients (including one from Group A), signs of decompensated liver disease were found with ascites. In three patients (including one from Group A), leukopenia (white blood cell count < 2000 /mm^3^) was found 14 days after the procedure. There were no deaths in the perioperative period or within 30 days after surgery.

## 4. Discussion

Irinotecan is a semi-synthetic analogue of the alkaloid camptothecin, which undergoes conversion in the liver parenchyma by carboxylesterases (CES-1 and CES-2) into the active metabolite 7-ethyl-10-hydroxy-camptothecin (SN-38) that inhibits DNA transcription. This activity of SN-38 is several hundred times greater than that of non-converted irinotecan. Most SN-38 is formed in the liver parenchyma from where it can diffuse into surrounding tumor cells [8]. Some studies have indicated a higher metabolism of irinotecan to its metabolite SN-38 in hepatocytes under the influence of carboxylesterase (CES-2) and a higher activity of this metabolite due to generated hypoxia and a decrease in pH [9]. In large tumor lesions, the diffusion of SN-38 may be impaired and the activity of CES-2 in tumor cells can be lower than in healthy liver parenchyma. In addition, extensive arterial vascularization of the tumor contributes to a greater washout of irinotecan/SN-38. This may lead to shorter and less exposure of tumor cells to SN-38 and a poorer therapeutic effect. Positron emission tomography CT studies have shown a significant association between areas of embolizate retention in CRC metastatic lesions and response to treatment. It has been shown that the active metabolite of irinotecan, SN-38, is more efficiently converted and released when the flow in the tumor vessels is stopped [10]. To obtain the retention of the embolizate covering the largest possible area of the lesion, it is necessary to at least temporarily stop the blood flow (stasis) in as many supply vessels as possible. This is facilitated by superselective embolizate administration, which allows the stasis to be limited to the tumor vessels and parts of the liver parenchyma around the tumor with which it is jointly vascularized, from a given sub- or segmental branch. The stasis generated during superselective administration of embolizate in the liver vasculature, by preventing rapid washout of irinotecan and its metabolite SN-38 from the tumor, may affect the treatment response. Furthermore, the embolization effect, caused by superselective microsphere administration, results in permanent occlusion of most of the pathological vessels in metastatic tumors. The use of the above results of stasis may contribute to a better therapeutic effect in patients in whom systemic chemotherapy has not had good effect. However, the presence of stasis may increase the incidence of adverse effects, mainly due to increased ischemia of the biliary plexuses [11]. In addition, stopping blood flow in branches of the hepatic artery may increase serum endothelial growth factor (VEGF) levels, which may contribute to tumor progression [12]. However, it has not yet been resolved whether hypoxia of the tumor is predominantly responsible for the increase in VEGF, or whether hypoxia of the peribiliary plexuses, which are subject to partial ischemia in each chemoembolization procedure, plays a greater role [13].

In our center, we initially administered embolizate at the level of the lobar arteries. With acquired experience, however, we noticed a frequent disparity between the response to treatment of small and larger tumor lesions in the same patients. While small lesions frequently showed a partial response (PR), for larger lesions we often noted only stabilization (SD) or even progression (PD) of the lesion. For this reason, we investigated a modification to the method of embolizate administration, to give, in the case of large metastatic lesions, the embolizate initially superselectively to the sub- and segmental arteries supplying the lesion in question with the creation of temporary stasis, and then the rest of the embolizate from the level of the lobar arteries without the creation of stasis. The results confirm our assumptions—the response to treatment in patients with large CRC-metastatic hepatic lesions depended on the selectivity of embolizate delivery in the hepatic artery branches and was found to differ between treatment groups. We demonstrated a greater radiologically determined response to treatment, with longer patient survival, with superselective embolizate administration. The current literature is dominated by arguments for lobar administration of irinotecan-loaded microspheres in order to deliver a chemotherapeutic agent to all liver lesions. However, arguments for superselective administration of chemoembolizate are being raised more frequently, especially in the case of large metastatic lesions, with the argument being that the healthy liver parenchyma would be less exposed to the toxic effects of irinotecan and that a higher dose can be delivered with a greater concentration of the active metabolite of irinotecan within the tumors [14]. In addition to the higher concentration of SN-38, the superselective administration of the embolizate allows the formation of stasis in the supply vessels, causing a greater devascularization effect of tumor foci. This is important because non-devascularized tumor foci are the most common site of recurrence in follow-up studies. Recent research has suggested a high response rate to stasis production and an acceptable level of adverse events [15].

The superselective administration of embolizate did not increase the incidence of complications, which with chemoembolization of colorectal metastatic lesions to the liver is estimated at about 1.6–7.2% [16], with a 30-day mortality of approximately 1.2% [17]. The reported complication rate in our study was low, which is consistent with conclusions from large multicenter studies regarding the safety of chemoembolization treatments and the impact on the number of significant complications. In our opinion, this makes it possible to safely use superselective embolization to achieve a better therapeutic effect for large CRC-metastatic liver lesions without affecting the number of significant complications. This is particularly important in patients who have previously received two or more palliative lines of systemic chemotherapy, who often develop tumor cell resistance to chemotherapeutic agents following low sensitivity of cancer stem cells towards regular cancer therapy [18,19]. Understanding the mechanisms of cancer drug resistance is critical to allow the development of effective treatments with sustained anti-tumor effects [20].

## 5. Conclusions

Superselective administration of chemoembolizate to vessels supplying large hepatic metastatic lesions from colorectal cancer gives a better response to treatment, and prolongs patient survival, without significantly increasing the risk related to the procedure.

## 6. Limitations

Our study was retrospective, without randomization of patients, and the study would have been better if the number of patients in each group was larger. All causes of death were included in the mortality analysis, without the ability to specify deaths due to cancer.

## Figures and Tables

**Figure 1 jpm-12-00414-f001:**
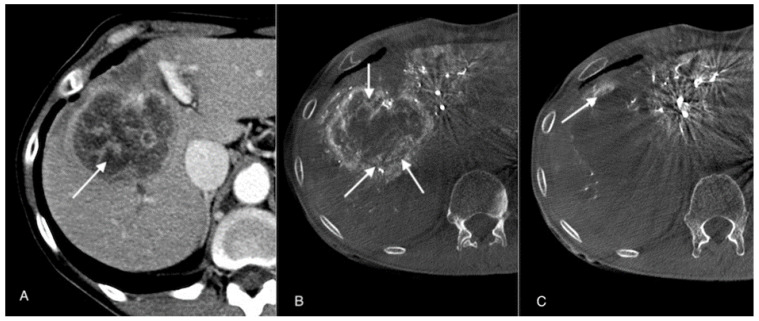
(**A**) Contrast-enhanced CT image before procedure: large metastatic lesion in the right lobe of the liver (arrow). (**B**) Cone-beam computed tomography image before the procedure: visible pathological tumor vessels (arrows). (**C**) Cone-beam computed tomography image three weeks after the procedure: almost complete devascularization of tumor vessels (arrow).

**Figure 2 jpm-12-00414-f002:**
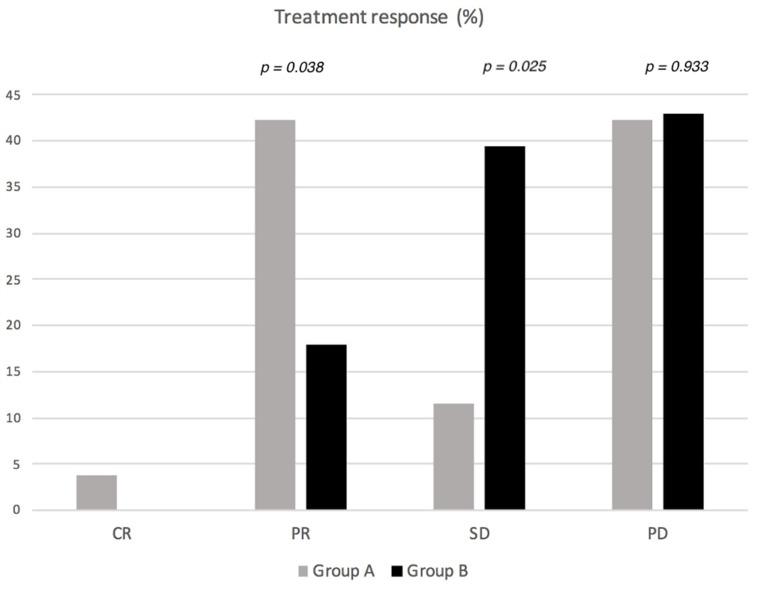
Comparison of treatment response between the two groups using chi-squared tests. Significance level was taken as *p* < 0.05.

**Figure 3 jpm-12-00414-f003:**
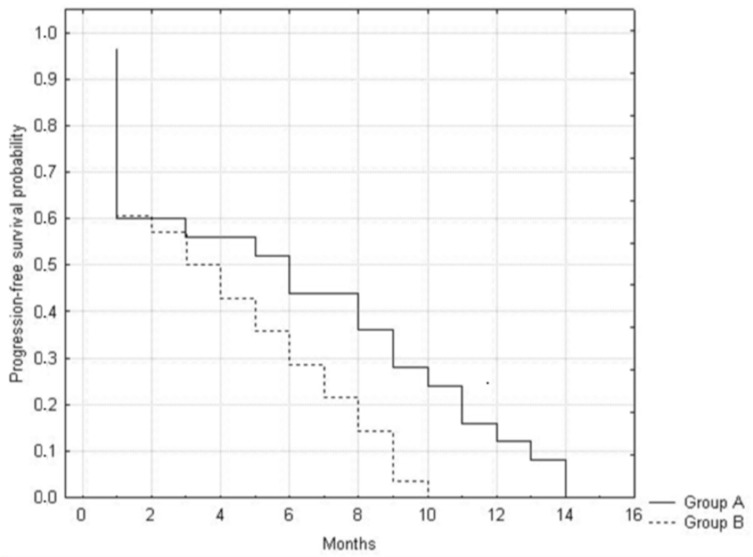
Kaplan–Meier progression-free survival analysis and log-rank test were performed to evaluate PFS in the two groups. *p* < 0.05 was considered significant.

**Figure 4 jpm-12-00414-f004:**
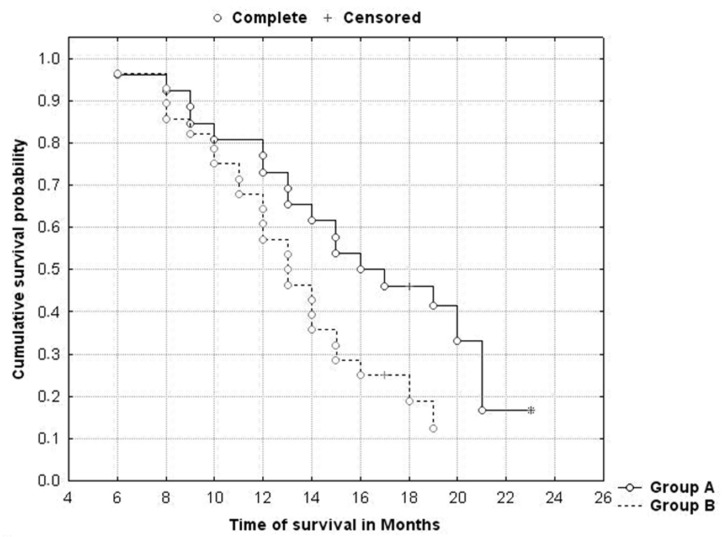
Kaplan–Meier survival analysis and long-rank test were performed to evaluate the OS between the two groups. *p* < 0.05 was considered significant.

**Table 1 jpm-12-00414-t001:** Technical details of therapy with drug-eluting microspheres (100 μm) loaded with irinotecan.

Parameter	Value
Total number of treatments	196
Number of treatments per patient: mean (range)	3.63 (2–4)
Number of treatments with each liver lobe:	
Right	100
Left	96
Number of treatments at each level of selectivity:	
Group A (Subsegmental/Segmental)	92
Group B (Lobar)	104

**Table 2 jpm-12-00414-t002:** Patient characteristics. Differences between the two groups were tested using *t* tests, Mann–Whitney U tests or chi-squared tests. A *p* value < 0.05 was considered significant.

Parameter	Group A (*n* = 26)	Group B (*n* = 28)	*p*-Value
Age, median (range)	68.3 (32–83)	66.5 (38–79)	0.103
Gender, female/male (*n*)	15/11	16/12	0.667
ECOG status: (*n*)			0.425
0	14	15	
1	12	13	
Tumor location: (*n*)			0.178
Bilobar	20	24	
Unilobar	6	4	
Number of liver metastases, median (range)	4.4 (1–10)	4.1(1–9)	0.339
Largest nodule size diameter, median (cm)	9.8	8.9	0.297
Extent of liver involvement (*n*, <25% left/>25% right)	21/5	23/5	0.201
Extrahepatic metastasis (*n*)	8	7	0.778
Number of prior systemic chemotherapy lines (median):	2.4	2.2	0.503
Prior liver surgery/ablation (*n*)	6/0	7/0	0.604
Prior locoregional therapy (*n*)	0	0	-

## Data Availability

The data presented in this study are available on request from the corresponding author.

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
