# Peer review of "Evaluation of CRC-Metastatic Hepatic Lesion Chemoembolization with Irinotecan-Loaded Microspheres, According to the Site of Embolization"

_jpm, 2022, doi:10.3390/jpm12030414_

Round 1

Reviewer 1 Report

Regarding the manuscript entitled “Evaluation of CRC-metastatic hepatic lesion chemoembolisation with Irinotecan-loaded microspheres, according to site of embolization”

This is a novel and well-written manuscript, but considering the following points would increase the quality of this paper:

  • Remove “.” At the end of the title
  • Try to make the abstract more to the point and easier to understand by improving its structure
  • The last sentence of your introduction needs to clearly reflect the aim of your study
  • Add the name of the manufacturer and its country in the material and methods section
  • The introduction section is weak. More updated information need to be added to this section.
  • Add more comparisons with other studies in the discussion section. Try to discuss the importance of your work in this section too
  • Adding an ethical code is required for this study
  • Also, the discussion section is weak. This paper has only 14 references, and most of them are old. The following papers could help you to increase the quality of your paper by adding updated information:

-Increasing the colon cancer cells sensitivity toward radiation therapy via application of Oct4–Sox2 complex decoy oligodeoxynucleotides

-Anticancer effect of X-Ray triggered methotrexate conjugated albumin coated bismuth sulfide nanoparticles on SW480 colon cancer cell line

-Anti-proliferative and Anti-metastatic Potential of High Molecular Weight Secretory Molecules from Probiotic Lactobacillus Reuteri Cell-Free Supernatant Against Human Colon Cancer Stem-Like Cells (HT29-ShE)

-Role of Oct4-Sox2 complex decoy oligodeoxynucleotides strategy on reverse epithelial to mesenchymal transition (EMT) induction in HT29-ShE encompassing enriched cancer stem-like cells

Author Response

Thank you  for the critiques and suggestions.

  1. Remove “.” At the end of the title.

Fixed in manuscript.

  1. Try to make the abstract more to the point and easier to understand by improving its structure.

Fixed in manuscript.

  1. The introduction section is weak. More updated information need to be added to this section. The introduction section is weak. More updated information need to be added to this section.

We have clarified the aim of our research work.

4.Add the name of the manufacturer and its country in the material and methods section.

Fixed in manuscript.

  1. Add more comparisons with other studies in the discussion section. Try to discuss the importance of your work in this section too. Also, the discussion section is weak. This paper has only 14 references, and most of them are old. The following papers could help you to increase the quality of your paper by adding updated information:

 In the discussion, we linked the latest research works related to the topic of our research. Due to the limited possibilities of the current drugs, we agree that molecular research into cancer cell resistance is of the greatest importance for the success of future therapies.

Reviewer 2 Report

The article presents the results of an original research aimed to evaluate chemembolisation of colorectal-cancer (CRC)-metastatic hepatic lesions with irinotecan-loaded microspheres.

The authors show that super-selective administration of chemoembolisation to vessels supplying large hepatic metastatic lesions from colorectal cancer gives a better response to treatment, and prolongs patient survival.

The manuscript is well structured. The topic is interesting, but as the authors point out, the sample is too small to draw generalizable conclusions. Further studies are needed.

In my opinion, the article can be published.

Author Response

Thank you for the critiques and suggestions.
In the discussion, we linked the latest research works related to the topic of our research.

Best regards
Marcin Szemitko

Reviewer 3 Report

Dear Authors, 

I read your article with interest. I have some comments: 

  • chemoembolization can be considered a palliative treatment, with low overall survival in patients with nonresectable hepatic disease. I think is necessary to explain better if you considered patients with unresectable diseases. Only 15 patients have an extrahepatic disease that can contraindicate resection, and the number/dimension/location of the lesions does not define the disease as nonresectable. So, I would suggest adding what do you consider resectable and more insight in your inclusion criteria
  • the follow-up at only 1 month is inadequate to evaluate the treatment response. Considering your cohort finished in 2019, I strongly suggest a more extensive follow-up, considering also the problem of vanishing lesions in the first months
  • would be interesting to know how many patients became resectable after the treatments

For these reasons, I don't consider at this moment the paper suitable for publication. 

Author Response

Thank you  for the critiques and suggestions.

1.Chemoembolization can be considered a palliative treatment, with low overall survival in patients with nonresectable hepatic disease. I think is necessary to explain better if you considered patients with unresectable diseases. Only 15 patients have an extrahepatic disease that can contraindicate resection, and the number/dimension/location of the lesions does not define the disease as nonresectable. So, I would suggest adding what do you consider resectable and more insight in your inclusion criteria

Each of the patients prior to TACE procedures was treated with palliative systemic chemotherapy with final failure. The patients were qualified for chemotherapy and re-qualified for TACE by the multidisciplinary oncological and surgical team, and was unsuitable for surgery, due to the localization of liver lesions and/or comorbidities.

2.Would be interesting to know how many patients became resectable after the treatments

Unfortunately, after treatment, none of the patients could be qualified for radical liver treatment. Only one patient achieved complete remission.

  1. The follow-up at only 1 month is inadequate to evaluate the treatment response. Considering your cohort finished in 2019, I strongly suggest a more extensive follow-up, considering also the problem of vanishing lesions in the first months

 We included a progression-free survival analysis in the follow-up assessment.

Most often, large metastases after TACE treatment do not decline in the first month. Post-treatment lesions tend to undergo mostly avascular necrosis, the size of which slowly decreases over time. This is shown in image 1c.

Round 2

Reviewer 1 Report

No comments

Author Response

Thank you  for the critiques and suggestions.

Reviewer 3 Report

Thank you for your replies. 

However, I still consider the paper not suitable for publication in this journal. I would expect more insight and explanation on the impact of this technique, especially the patient selection and who would benefit most from this procedure. 

Patients are not considered suitable for any other treatments because of liver cancer localization and comorbidities. However, looking at your exclusion criteria (ECOG >=2, liver failure, ascites, bilirubin level >3 mg/dl, involvement of more than 50% of liver parenchyma, renal failure (creatinine >2 mg/dl), 106 thrombocytopenia (platelets <50 000/mcl), allergy to contrast) it is not clear the decision to perform the procedure rather than other possible treatments. 

In the treatment of CRLMs patient selection is the key for an improved patient outcome, as offering this type of procedure means offering almost a palliative care compared to other possible treatments (liver resection, ablation, and in highly selected case liver transplant, with survival at 5 years almost at 70%). 

Author Response

Dear Editor.

Thank you  for the critiques and suggestions.

Initially the multidisciplinary oncology and oncosurgical team had rejected each of these patients from surgery, due to the contraindications  (tumours localization, bilobar disease, multiple lesions, large tumours, extrahepatic disease ) or comorbidities and qualified these patients for  palliative chemotherapy.

All patients progressed after FOLFIRI and most of them after FOLFOX (some were intolerant to oxaliplatin).   Next, the multidisciplinary team re-qualified these patients for TACE.

Chemoembolization was one of the very few palliative treatment options for these patients.

We specified in the manuscript that the study concerns palliative-treated patients, with inoperable  lesions and the superselective embolization should be used for large metastatic CRC lesions to the liver, resistant to chemotherapy (line 211-212; 599-604).

Round 3

Reviewer 3 Report

Dear Authors, 

Thank you for your revised paper. 

This manuscript is a resubmission of an earlier submission. The following is a list of the peer review reports and author responses from that submission.